# Influence of Haptoglobin Polymorphism on Stroke in Sickle Cell Disease Patients

**DOI:** 10.3390/genes13010144

**Published:** 2022-01-14

**Authors:** Olivia Edwards, Alicia Burris, Josh Lua, Diana J. Wilkie, Miriam O. Ezenwa, Sylvain Doré

**Affiliations:** 1Department of Anesthesiology, Center for Translational Research in Neurodegenerative Disease, University of Florida College of Medicine, Gainesville, FL 32610, USA; oliviaedwards@ufl.edu (O.E.); aliciaburris@ufl.edu (A.B.); joshlua@ufl.edu (J.L.); 2Department of Biobehavioral Nursing Science, University of Florida College of Nursing, Gainesville, FL 32610, USA; diwilkie@ufl.edu (D.J.W.); moezenwa@ufl.edu (M.O.E.); 3Departments of Neurology, Psychiatry, Pharmaceutics, and Neuroscience, McKnight Brain Institute, University of Florida College of Medicine, Gainesville, FL 32610, USA

**Keywords:** brain ischemia, genotype, hemolytic anemia, hospitalization, inflammation, mini-stroke, oxidative stress, pain crisis, silent cerebral infarction, therapy, vaso-occlusion

## Abstract

This review outlines the current clinical research investigating how the haptoglobin (Hp) genetic polymorphism and stroke occurrence are implicated in sickle cell disease (SCD) pathophysiology. Hp is a blood serum glycoprotein responsible for binding and removing toxic free hemoglobin from the vasculature. The role of Hp in patients with SCD is critical in combating blood toxicity, inflammation, oxidative stress, and even stroke. Ischemic stroke occurs when a blocked vessel decreases oxygen delivery in the blood to cerebral tissue and is commonly associated with SCD. Due to the malformed red blood cells of sickle hemoglobin S, blockage of blood flow is much more prevalent in patients with SCD. This review is the first to evaluate the role of the Hp polymorphism in the incidence of stroke in patients with SCD. Overall, the data compiled in this review suggest that further studies should be conducted to reveal and evaluate potential clinical advancements for gene therapy and Hp infusions.

## 1. Introduction

### 1.1. Sickle Cell Disease

Sickle cell disease (SCD) is a group of autosomal recessive disorders that affect an estimated 20 to 25 million people worldwide [1], making it the most prevalent monogenic disorder and a serious public health concern. Inheritance of this disorder is concentrated in sub-Saharan African, South Asian, Middle Eastern, and Mediterranean regions [2,3]. Due to the recessive-trait nature of SCD, even larger populations carry sickle cell trait, maintaining the chances that SCD will be passed on through generations. Every year, an estimated 300,000 infants are born with SCD worldwide, adding to the existing millions of patients seeking treatment [2]. SCD is a hemolytic disorder caused by a range of mutations in the gene responsible for coding the β-globin (HBB) subunits of hemoglobin (Hb). These mutations usually result in abnormal versions of Hb, which are capable of polymerizing, leading to malformed, sickle-shaped red blood cells (RBCs). SCD is a form of hemolytic anemia, a class of disorders characterized by high rates of hemolysis. As these deformed and rigid RBCs travel through the vasculature and supply major organs, high levels of hemolysis occur, triggering signaling pathways that can lead to oxidative stress, free radical formation, chronic pain crises, and other detrimental symptoms that are characteristic of SCD. The resulting byproducts of hemolysis are of particular concern due to their high natural toxicity to the body’s life-sustaining organs. Free heme, a direct product of Hb destruction, is the main culprit of many symptoms associated with SCD. Among other detriments, the circulation of toxic free heme can cause vaso-occlusion-related pain crises, vasoconstriction, proinflammatory macrophage signaling and cytokine release, acute kidney injury, and silent cerebral infarction (SCI) [4]. These symptoms can aggravate existing SCD symptoms and worsen the independent quality of life of those suffering from chronic SCD. Widespread vasoconstriction exacerbates vaso-occlusion and resulting pain crises by further constricting existing blood flow blockages, in turn increasing premature hemolysis rates, worsening the severity of SCD pathophysiology, and increasing the risk of damaging critical organs. SCD is the umbrella term used to group the many subtypes of this monogenic disease because there are multiple possible Hb genotypic variations. Throughout this review, we will focus on the most severe and commonly diagnosed form of SCD: sickle cell anemia (SCA). SCA is characterized by a double inheritance of the HbS genetic mutation, resulting in a homozygous sickle cell allele genotype (HbSS). Expression of HbS results in the production of malformed sickle-shaped RBCs prone to causing excessive vaso-occlusive crises (VOCs) due to their characteristic sharp edges catching on walls of connective tissue within the circulatory system.

It is common for patients with SCD to experience chronic pain crises associated with VOC. The deformed RBCs stick to the endothelium and collect to form clots throughout the body, triggering premature hemolysis and anti-inflammatory response pathways at an exponentially increased rate. When these VOCs occur specifically within the cerebral vasculature, stroke occurrence also increases. These strokes are often silent or mini-strokes that do not present symptomatically. Alternatively, they can be overt ischemic strokes, causing severe, lasting symptoms in patients. SCD and stroke incidence are highly associated, especially in children, who have been shown to be 221 times more likely to experience stroke, which is 410 times more likely to be classified as cerebral infarction [5]. Stroke is of special interest due to the often irreversible damage it causes to the nervous system. This damage is particularly difficult to manage if the patient has pre-existing chronic conditions such as SCD. Thus, there is a clear rationale for studying the range in SCD symptoms and outcomes, particularly concerning genetic variations in this patient population, in hopes of designing more specialized treatment plans and palliative care. This review aims to identify the influence of the haptoglobin (Hp) genetic polymorphism on clinical manifestations of SCD.

### 1.2. Haptoglobin Polymorphism

Hp is a glycoprotein mainly synthesized by the liver in response to increased inflammatory signaling associated with hemolysis [6]. Hp primarily functions to remove toxic, free, oxygen-loaded Hb from circulation to prevent the formation of free radicals and subsequent oxidative tissue damage. Hp binds cell-free Hb with high affinity and is later taken up by macrophages and monocytes via the receptor CD163 (Figure 1). This functional ability has been found to rely heavily on the inherited genotype of an individual and is particularly relevant to SCD pathophysiology. The Hp gene is coded on chromosome 16 (16q22) by two wholly expressed alleles, HP 1 and HP 2, resulting in the exhibition of three possible genotypes in humans: Hp 1-1, Hp 2-1, and Hp 2-2 [7]. The HP 1 allele is unique in its expression as either fast (HP 1F) or slow (HP 1S), referencing its relative speeds dependent on the inherited α-chain type [6]. This HP 1 amino acid variant provides six detailed genotypes: Hp 1S-1F, Hp 1S-1F, Hp 1F-1F, Hp 2-1S, Hp 2-1F, and Hp 2-2 [7]. Literature suggests that the HP 1 allele may be biologically advantageous due to its shorter α-chain, resulting in smaller size and quicker exportation of Hb and increased antioxidative abilities [6,8]. In contrast, the HP 2 allele has been correlated with a lower blood concentration of Hp and a decreased ability to bind to Hb [9]. Previous literature has provided strong evidence that a Hp 2-2 genotype increases the risk of cardiovascular events such as stroke in patients with diabetes [10], which raises the question of how the Hp genotype affects stroke incidence in other populations susceptible to stroke, such as patients with SCD. Our group has also previously reviewed the role of Hp polymorphism in subarachnoid hemorrhage (SAH). We found strong evidence that Hp 1-1 can be taken up by cells faster, as well as cross the blood–brain barrier, which may ameliorate secondary injury after SAH [11]. However, it should be noted that we did not previously distinguish between the HP 1S and HP 1F alleles. The rationale for this literature review relies on the premise that SCD, a form of hemolytic anemia, has previously shown significant trends toward Hp concentration and activity regarding symptom severity [12,13].

Recent literature has brought to light several variant forms of SCD due to heterozygous inheritance of Hb abnormalities, such as Hb SC, Hb SD, Hb SE, and Hb Sβ^+^-thalassemia [14]. While there is a significant overlap between these diseases, Hb SS is both the more common form in the US [15,16] and the more severe form in terms of hemolysis. Thus, we have opted to focus solely on the homozygous Hb SS form of SCD for the purposes of this review.

### 1.3. Critical Role of Haptoglobin

RBCs contain high levels of Hb tetramers, which are released into the body after hemolysis. A single RBC has been shown to contain at least 250 million Hb molecules, which upon lysis become free in the blood [11]. Free Hb can then travel in its broken-down dimeric form, allowing quicker transport to the kidneys, where Hb is highly toxic. In normal hemolysis, Hp molecules will bind to free Hb dimers and form Hb-Hp complexes, which are taken to the liver for breakdown and excretion. In SCD, however, hemolysis occurs at a much higher rate that is often too much for the body to handle. As a result, free oxidized Hb escapes excretion and remains within the circulation. Free oxidized hemoglobin (met-hemoglobin) and heme, byproducts of hemolysis, are ligands of the Toll-like receptors (TLRs). When these molecules are not bound to Hp after RBC rupture, they can bind to TLRs, which can trigger the nuclear factor kappa B (NFκB) transcription factor. Signaling NFκB activity leads to increased expression of proinflammatory genes that code for a variety of immune responders, including inflammatory cytokines, immature neutrophils that can increase production of reactive oxygen species (ROS), and other cycles that promote large organ dysfunction [17]. This futile cycle of proinflammatory response may explain the chronic pain crises commonly experienced by patients with SCD. NFκB becomes especially important in the event of ischemic stroke because proinflammatory molecules can further damage the affected tissue and cause cell death [18]. Hp has, therefore, been identified as a probable key identifier of chronic disease progression of SCD. For example, a biologically reduced concentration or functionality of Hp could severely impact the body’s ability to cause a large enough immune response to manage the increased rates of hemolysis caused by SCD. Hp is also known to play an antioxidative role in preventing oxidative tissue stress. Cell-free Hb is broken down into heme, iron, and globin. The most relevant major downstream pathway of free heme is free radical formation via the Fenton iron reaction; these free radicals interact with iron to produce ROS, which are harmful to the surrounding cells and tissue. In addition to this component that causes oxidative stress, free heme has the added effect of scavenging available nitric oxide (NO), an important upregulating molecule for vasodilation [19]. The scavenging and consequential depletion of NO in the bloodstream lead to vasoconstriction, which can exacerbate the natural pathophysiology of SCD and stroke [20]. 

This complex cascade of harmful reactions can be prevented by the correct interplay of the Hp–Hb binding mechanism. However, in patients diagnosed with a hemolytic anemia disorder, this mechanism is often inadequate. A hemolytic patient’s supply of free unbound Hp is constantly being depleted and cannot match the body’s increased demand resulting from the above-average hemolysis rates. This concept has been corroborated by previous findings that link Hp and hemopexin depletion with Hb-mediated oxidative stress [13]. Toxic species can escape removal from the organs and bloodstream and cause serious damage to critical major organs, specifically the kidneys. Introducing the significance of the Hp polymorphism and its subsequent phenotypes into this discussion only supports the need for investigating the different roles of Hp, their affinities to the various Hb variants, and its consequences on vaso-occlusion and free radical damage and inflammation. If current and future clinical trials can identify a more biochemically active and clinically advantageous Hp allele (HP 1F, HP 1S, HP 2) with consistent significance, we may be able to predict and prevent SCD disease progression more accurately via new therapies such as Hp infusions. Preclinical models of Hp infusion have also shown a linked reduction in kidney damage, oxidative stress, and vascular injury, all of which are major complications of SCD in humans [21]. 

### 1.4. Blood Exchange Transfusions

One of the most common therapeutic agents available to chronic and acute cases of SCD is blood exchange transfusion therapy. Traditional blood transfusions consist of transferring nutrient-rich blood from a healthy donor into the veins of a naturally deficient patient, usually to supply nutrients to a patient who is not capable of producing or maintaining proper nutrient levels in the circulatory system. In patients with SCD, blood exchange transfusions aid in supplying healthy hemoglobin (HbA) and normal biconcave RBCs, increasing the oxygen-carrying capacity of the blood and reducing the complications and likelihood of VOC [22]. In essence, these transfusions can temporarily reverse the effect of the mutated gene responsible for producing HbS by keeping the levels of HbS below 30%; maintaining this balance of HbA and HbS has been shown to reduce symptoms of anemia in patients with SCD [23]. In addition, patients who receive transfusions often have reduced symptoms of anemia because the transfused blood volume compensates for the decreased blood viscosity associated with SCD [8]. Stroke occurrence is also reduced in patients with SCD who regularly receive blood exchange transfusions. Consistent participation in long-term blood exchange transfusions tends to decrease the severity of pain crises, likely related to decreased VOC. We previously discussed how blood transfusion should be minimized because of side effects and should be part of a formal hospital blood management program [24]. Such transfusion, when necessary, also tends to slow disease progression compared to patients with little or no history of blood exchange transfusion treatment [23]. This is an easily predicted outcome because infusing healthy Hb should decrease the proportion of HbS relative to infused HbA, reducing the concentration of sickle-shaped RBCs that implicate VOCs. With reduced VOC frequency, we can expect a reduced incidence of cerebral vessel blood clots and, therefore, fewer ischemic stroke events. This information raises the question of whether we should begin screening blood donors for their respective Hp genotypes to best care for patients with SCD and decrease and mitigate the negative side effects of blood exchange transfusions. 

## 2. Methods

The papers used in this review were retrieved through extensive searches in several search engines, including Google Scholar, Embase, OneSearch, and Primo (at the University of Florida’s library database). Articles that were relevant to the primary focus of haptoglobin, sickle cell disease, and stroke dating from 2010 to the present were included. The following search terms were used in all databases: “sickle cell anemia, stroke and haptoglobin”, “(‘sickle cell’) AND (‘haptoglobin’ OR ‘Hp ’) AND (‘stroke’ OR ‘CVA’ OR ‘vasospasm’)”, “haptoglobin, brain ischemia, and sickle cell anemia”, “(‘haptoglobin’ OR ‘polymorphism’) AND ‘stroke’/exp AND (‘sickle cell anemia’ OR ‘sickle cell disease’)”, “haptoglobin’ AND ‘vasoconstriction’ AND (‘sickle cell anemia’ OR ‘sickle cell disease’), “cerebral ischemia, haptoglobin and sickle cell disease”. In addition, abbreviations were used in place of sickle cell disease (SCD), sickle cell anemia (SCA), and haptoglobin (Hp) to perform a more comprehensive search. Each author searched independently to mitigate potential bias, and the list of resulting papers was cross-referenced. The last search was completed on 13 November 2021. 

## 3. Current Evidence Regarding Hp Genotypes and Stroke

Among the first to study the role of Hp genotype in SCD, Atkinson et al. showed how the Hp polymorphism might acutely affect the hemolytic response [9]. When children living in malarial regions were surveyed by their Hp group, a notable increase in Hb clearance was observed in children with double HP 2 inheritance, meaning that the HP 2 allele was correlated with an increased risk for anemia [9]. Although there is overwhelming evidence that there is no association between Hp genotype and SCD, we are still interested in collecting data on how the Hp genotype may influence SCD outcomes because this disease can manifest in a variety of unique ways [9,25]. From the literature we reviewed, we observed that most clinical outcomes were insignificantly correlated with the Hp genotype. The relationship of the Hp genotype to stroke incidence found by Barbosa et al. suggests that Hp 1F-1F may be advantageous due to the correlation with lower stroke incidence [8]. In contrast, Olatunya et al. reported no remarkable trends of a similar nature [26]. As previously mentioned, studies on Hp have theorized that the HP 2 allele predicts a poorer prognosis for outstanding conditions, including SCD [27]. Adekile and Haider identified some clinical significance regarding the HP 2 allele; in the Kuwaiti population with SCD, HP 2 dominance was correlated with higher hemolysis rates than in the South Nigerian population [28]. This finding contradicts the original premise of this clinical study, which selected these populations for their SCD clinical severity. According to the assumption that South Nigerian patients with SCD present with more severe cases, higher hemolysis rates and dominance of the “inferior” HP 2 allele should have been observed in the South Nigerian population. In fact, the Kuwaiti populations showed the highest inheritance of the HP 2 allele and frequent VOC, but when stratified, no correlation was found between variables. The authors of this study noted that the number of insignificant results may have been caused by the small sample sizes used to generalize large regional populations, which may also have a skewed representation of genotypic variation [28]. Only one of these clinical studies, Barbosa et al., conducted polymerase chain reaction (PCR) runs to differentiate the two subtypes of the HP 1 allele, HP 1F and HP 1S, delineating relative size and speed. To fully understand how Hp polymorphism affects SCD and stroke occurrence, we must expand our knowledge on the implications of this mutation alone [8]. From our search, we found 5 articles that evaluated stroke incidence/outcomes and Hp phenotype (Table 1), although it should be noted that none of the articles specifically investigated the relationship between the two and study them only tangentially.

Adekile and Haider produced a unique study to identify the significance of Hp polymorphism in patients with SCD from Kuwait and South Nigeria, using respective control groups for comparison [28]. These populations were chosen for their previously studied SCD complications; Kuwaiti patients typically display milder SCD subphenotypes than South Nigerian patients’ more chronic, severe subphenotypes. Both the Kuwaiti and the South Nigerian populations showed insignificant Hp genotype intragroup distribution (*p* = 0.78, *p* = 0.41) but statistically significant intergroup distribution between the two SCD groups (χ^2^ = 31.4, *p* < 0.01). The lack of significant intragroup HP allele distribution supports the theory that Hp genotypes are heavily influenced by ethnicity and patterns of geographical spreading. No significant distribution regarding HP 2 inheritance and VOC frequency among the Kuwaiti group was found; Kuwaiti patients with SCD demonstrated frequent VOCs with low stroke incidence, which the authors concluded was more significantly correlated to other genetic markers (AI β haplotype and higher fetal Hb levels) than Hp genotype. According to the literature, the HP 1 allele was predominant in the South Nigerian SCD group, which should have predicted higher hemolysis rates due to the more efficient Hb-binding ability of HP 1. This study showed that the opposite was true, with the Kuwaiti HP 2-dominant patients with SCD showing higher levels of hemolysis [28]. This study did not stratify results for the South Nigerian SCD group as it did for the Kuwaiti SCD group, and it used a narrow subject population.

Cox et al., one of the first groups that sought to connect Hp genotype with stroke in patients with SCD, found inconclusive results that contradicted their hypothesis regarding Hp [29]. They conducted a study in a population of healthy pediatric patients with SCD in Tanzania and focused on measuring cerebral blood flow (CBF) using transcranial Doppler (TCD). Their reasoning behind the study was that they would be able to assess how CBF in patients with SCD was affected by three polymorphisms: glucose-6-phosphate dehydrogenase (G6PD), α-thalassemia, and the Hp genotype. The study provided evidence reinforcing the theory that α-thalassemia has a significant effect on CBF and, subsequently, stroke; however, it also failed to find any significant differences between the genotypes of the other two polymorphisms, G6PD and Hp, in terms of CBF [29]. These negative results may be related to the researchers’ decision to exclude patients with a history of stroke, recent transfusion, or recent manifestation of SCD. By excluding the more severe cases of SCD, it is possible that the authors inadvertently obscured the effects of Hp genotype. The authors also stated that any adverse effects of the Hp 2-2 genotype may either be overwhelmed by the sheer amount of cell-free Hb that is present in the disease, compensated for by heme oxygenase 1, or some combination of the two factors. Either way, more research is needed to address whether the Hp genotype can cause increased vasoconstriction and subsequently worse outcomes during stroke.

While Cox et al. focused on the occlusion and perfusion aspects of SCD in Tanzania [29], Barbosa et al. conducted a similar cross-sectional study in the Brazilian population focused on the iron overload aspect of SCD [8]. Researchers conducted a cross-sectional clinical study using a convenience sample of 78 Brazilian patients with HbSS SCA with the goal of identifying significant differences between those with normal iron levels and acquired iron overload in relation to their respective Hp genotypes. This population in Federal District, Brazil, was selected for its ethnic diversity to eliminate extraneous independent variables capable of influencing genetic inheritance. Acquired secondary iron overload is common in patients with hemolytic anemia disorders due to frequent blood exchange transfusions. The authors concluded that there was no statistically significant distribution of Hp genotypes among patients with SCA with and without iron overload. Hp 1F-1S and Hp 1F-1F were observed in only 15% of the participants, possibly suggesting that the HP 1 allele is less common among patients with SCA. There was a significantly higher stroke incidence in patients with Hp 1S-2 than predicted by the Hardy–Weinberg equilibrium (HWE) equation used prior to conducting the study (*p* = 0.005). Contrary to outcomes predicted with the HWE, patients with Hp 1S-2 had the highest hospitalization for stroke (OR = 6.346, *p* = 0.005) and stroke sequelae (OR = 6.556, *p* = 0.005). Conversely, Hp 1F-1F patients had the lowest hospitalization for stroke and stroke sequelae, which was less than that predicted by HWE using the notion that the smaller size of the HP 1 allele should allow quicker entrance into the interstitial fluid, reducing seizure recurrence. No statistical comparison was provided between ages due to inconsistent biological values obtained and only two participants in the ≥60 age range. It is also important to note that patients had varied and inconsistent treatment histories for their condition. Out of the 78 patients, 41 had a history of blood exchange transfusion therapy. In concurrence with this review, this study discussed the practical clinical advantages of exchange transfusions regarding decreased stroke occurrence; therefore, the observed correlation between increased transfusions and decreased stroke was expected, independent of the Hp genotype. These results should not be generalized to the whole area population because this study was conducted using a convenience sample, which creates a high risk for underrepresentation and overrepresentation of genetic inheritance and SCD complexities [8].

Olatunya et al**.** conducted the most recent cross-sectional clinical study associating genotypic variation with SCA phenotypes [26]. The authors of the study aimed to identify any significant distribution of Hp genotypes between 101 young (2–21 years old) Nigerian patients with SCA and 64 healthy Nigerian control patients. The authors reported that the HP 1 allele had the highest inheritance among patients with SCA, occurring in approximately 62%, but the control group showed even higher inheritance, at approximately 73%. The high representation of the HP 1 allele in this patient population opposes the previous findings from Barbosa et al. Still, it contributes to the existing theory that Hp allele distribution is influenced more by ethnic origin than SCA diagnosis. The authors also found no significant relationship (*p* = 0.375) between stroke incidence and any Hp genotype. Only five (4.9%) of the 101 patients with SCA had a history of hospitalization for stroke; two were Hp 1-1, one was Hp 2-1, and two were Hp 2-2. This low incidence of stroke should not be generalized to the entire SCA population because the oldest patient in this study was 21 years old, although it should be noted that pediatric patients with SCD are at a higher risk for stroke, with an estimated 11% of patients with SCD experiencing stroke before age 20. Furthermore, Hp 2-2 showed an insignificant (*p* = 0.922) correlation to increased VOC, which had been previously characterized by the lower antioxidant and anti-inflammatory capabilities of the HP 2 allele. From this study, the authors concluded that the Hp genotype is likely not a reliable indicator of SCA clinical severity and claimed to disprove existing literature suggesting that specific Hp genotypes, namely Hp 2-2, influence poorer clinical outcomes. A notable theory mentioned in this study is that Northeast Brazilian populations may show similar genotyping as Nigerian populations due to the slave trade in previous centuries. Although the aforementioned study by Barbosa et al. did not show an equivalent distribution of Hp alleles as this study did, an earlier study specifically within Northeastern Brazil did [8,26,30]. This difference may imply that Hp genotypes among general populations are less predictable than previously theorized.

## 4. Discussion

Although controversial, there is evidence showing that HP 2 alleles are disadvantageous in SCD. Barbosa et al., who conducted one of the few studies implicating Hp 2 alleles in stroke occurrence, also conducted the only study covered that distinguished between Hp 1F and Hp 1S. Indeed, the lack of association between stroke and Hp genotype found by studies such as those of Adekile and Haidar, Cox et al., and Olatunya et al. may be explained by their lack of distinction between the two subtypes of HP 1. Regardless, current literature has made it clear that Hp genotype is not the deciding factor in the severity of SCD. However, because Quimby et al. demonstrated that Hp infusion was feasible in the context of addressing SCD symptoms [21], researchers should still consider the therapeutic potential in the identification and selective use of certain Hp phenotypes. Although a meta-analysis of current literature would provide more conclusive evidence of an association, it is difficult to conduct one because the existing research varies too widely in design.

There is limited and conflicting evidence on whether these proinflammatory cytokine levels are influenced by Hp polymorphism. The literature has repeatedly demonstrated the correlation between SCA and increased expression of proinflammatory cytokines, even during periods without crisis [31,32,33,34]. Tumor necrosis factor (TNF) is an important proinflammatory cytokine responsible for a complex downstream signaling pathway of inflammatory molecules and transcription factors, including the aforementioned NFκB. Levels of TNF are significantly (*p* = 0.006) increased in patients with steady-state SCA when compared to healthy controls, suggesting levels may be even higher when in crisis [34]. Interleukin (IL)-1 and IL-8 induce similar effects on the body’s immune system, releasing proinflammatory molecules to regions of stress. These particular ILs were significantly observed in concentrations as high as four times greater in patients with SCD than healthy controls [25,31,33,34]. Overexpression of such molecules can lead to added pain crises, even when a patient’s SCD symptoms are well managed. Notably, increased expression of these cytokines causes a loss of endothelial structural integrity and expression of adhesion molecules. These events can cause sickled RBCs and leukocytes to become sticky, increasing the risk of endothelial injury, vessel blockages, hemolysis, and ultimately VOC-related pain [25]. One clinical study presented evidence proposing a significant relationship between Hp polymorphism and modulation of anti-inflammatory IL-6 and IL-10 production. Guetta et al. observed an increased ability to downregulate inflammation via IL-6 and IL-10 in patients with SCD with the Hp 1-1 genotype, possibly providing an explanation for these patients’ associated lower levels of TNF [25,35]. In a more recent study, no significant association was found. It is possible that with more clinical investigations, we will further understand the intense upregulation of proinflammatory responses in patients with SCD and whether Hp polymorphism plays a major role. Such an understanding may better inform our research in designing individualized and targeted therapy for this patient population.

Patients with SCD are at higher risk of stroke for a few reasons. With a decreased ability for Hp to bind with all free Hb upon chronic hemolysis, there is an increased rate of NO scavenging by the cell-free Hb in circulation. As previously mentioned, patients with SCD often have uncontrolled ROS overproduction as a result of cell-free Hb circulation. When in the presence of NO, ROS undergoes redox reactions to form reactive nitrogen species, further harming the body by scavenging NO and producing toxic species [36]. With this severe depletion of circulating NO, which is one of the body’s most crucial vasodilators, widespread vasospasm and, subsequently, insufficient circulation of oxygenated blood is observed. When these vasospasms occur within the cerebral vasculature, the patient is at increased risk of silent and ischemic stroke. In addition to increasing stroke risk, vasospasms create an environment prone to subsequent VOC. Vessels narrow and sickle RBCs catch on endothelial tissue and form more clots, instigating more hemolysis and, therefore, more NO scavenging. This ultimately amplifies the existing high risk for ischemic stroke (Figure 2) [11]. Hp genotype may play a significant role in the severity of this cycle experienced by patients with chronic SCD. Preeclampsia and type 2 diabetes are acquired conditions characterized by increased systolic blood pressure, which may be caused by recurrent vasoconstriction possibly related to NO scavenging. There have been clinical studies on both conditions to identify a predominant Hp allele among patients with the lowest levels of NO and highest levels of vasoconstriction. Both studies observed that single and double inheritance of the HP 2 allele on chromosome 16 was significantly correlated with increased vascular tone due to greater apparent NO depletion [37,38,39]. Conducting a similar study using a population of patients with SCD of varying Hp genotypes may provide an important conclusion informing Hp genotype-specific infusion design. With the current technology available, however, the best future direction for researchers interested in this topic is to give blood exchange transfusions using blood with the Hp 2-2 or the Hp 1-1 phenotype to patients with SCD and compare its ability to attenuate the effects of the disease. Better yet, researchers should also consider distinguishing Hp 1S and Hp 1F when designing such a study.

The single most reliable predictor of overt ischemic stroke is evidence of recurrent SCI. SCI is commonly defined as symptomatically covert incidents of ischemia causing lesions visible via magnetic resonance imaging in patients with SCD. These occur in children with SCD younger than age 6 and increase in occurrence with age. A study showed that 27% of children with SCD younger than age 6 experienced SCI and by the age of 14, 37% of children with SCD experienced SCI [40]. A similar study in an adult population of patients with SCD with a relatively low mean age of 30 years showed that 53% had brain lesions consistent with SCI [41]. Understanding risk factors and genetic predispositions for SCI in patients with SCD seems crucial to prevent overt ischemic stroke, which can irreversibly alter neurological functions and cause severe cognitive deficits as early as elementary age. It has been shown that a history of seizure, low Hb count, and high blood pressure are major risk factors for SCI occurrence [42]. It is a probable assumption that depletion of NO stores causing peripheral vasoconstriction may affect blood pressure, possibly encouraging SCI incidence. Our search found no results for clinical studies correlating Hp genotype and SCI occurrence, which may be beneficial in understanding how SCD biomarkers impact stroke.

As mentioned earlier, this review focused solely on the homozygous Hb SS form of SCD. However, it should be noted that Hb SC has been found to be the prevalent form of SCD in a study from Burkina Faso [43], whereas Hb Sβ-thalassemia was found to be the prevalent form in Greece [44], as well as a small cohort from India [45]. The findings from this review may not be as applicable to these areas due to the unique characteristics of each type of SCD, but there remains a considerable gap in the literature regarding these diseases.

### Future Directions

There are few safe, long-term, and effective treatment options for SCD. Daily hydroxyurea treatment and regular blood exchange transfusions are two of the most commonly prescribed therapeutic treatments to manage life-altering and often debilitating symptoms, especially chronic pain crises. Although these two treatments have been shown to improve chronic and acute SCD, both require strict adherence, limiting the scope of patients who can be treated via these methods. In addition to the recommended routine adherence, regular blood exchange transfusions are costly and can put patients with SCD at further risk for disease progression by causing iron overload [22,40]. ROS are produced by excess iron via the Fenton reaction, exacerbating SCD symptoms of blood component toxicity [7]. A single-center study observing patients with chronic SCD who received regular RBC exchange transfusions reported that 45% of patients experienced complications or discomfort from treatment [22]. Beyond basic complaints, children placed on regular blood exchange transfusions to prevent recurrence of overt stroke remained susceptible to SCI and secondary overt stroke at significant rates [46]. Regardless, as the life expectancy for patients with SCD increases [47], so too will the prevalence of the disease. SCD will increasingly affect patients worldwide, creating a growing need for accessible and preventative treatments. We believe that exchange transfusion and hydroxyurea therapy are the optimal current options. However, Hp infusions may be a future option to manage SCD symptoms, serving as a complement rather than a replacement to regular blood transfusions. Indeed, the clinical use of Hp is not a distant possibility considering that CSL Behring is conducting a clinical trial on the use of hemopexin, which they designated as CSL889 (ClinicalTrials.gov identifier: NCT04285827). Although further study is needed, we theorize that the use of clinical-grade Hp can lead to decreased pain crises and overall increased quality of life in patients with SCD.

Exchange transfusion has strengths and limitations. Researchers should also take into consideration the latest developments in SCD therapy and Hp research. The US FDA recently approved two new drugs for SCD treatment, voxelotor and crizanlizumab [48]. Crizanlizumab helps ameliorate and decrease VOC by inhibiting P-selectin, thereby decreasing RBC adhesiveness [49]. Crizanlizumab and Hp act on different pathways, so theoretically the two treatments should be compatible, although no studies have been conducted corroborating this relationship. Voxelotor, on the other hand, has more relevance to Hp because both molecules bind to Hb. Unlike Hp, however, the drug actively prevents the polymerization of HbS, thereby decreasing deformation of RBCs [50]. The binding site for voxelotor, which is on the N-terminus of the α-chain of Hb, differs from the binding sites of Hp. Theoretically, the two molecules should not compete in the binding of Hb, but more investigation is needed to determine whether the drug is compatible with Hp infusion.

We suggest several future directions of research for generating Hp for therapeutic use. The first and best-characterized option would be the purification of Hp from human plasma. There have been several published methods for doing so, including affinity chromatography [51], ion-exchange chromatography [52], and tangential flow filtration [53]. Recombinant Hp production has also been performed in FS293F cell culture [54]. Still, its applicability to mass production is limited by the particular conditions needed for post-translational modification of Hp. The final option that we suggest is the upregulation of endogenous Hp production. RNA therapy, specifically mRNA therapy, has recently seen interest after the development of mRNA vaccines. Considering that it is well within our capabilities to produce viral proteins, it should be equally possible to do so with an endogenous protein such as Hp, although no such studies have been conducted to date.

## 5. Conclusions

This review was conducted to elucidate a connection between SCD, Hp, and stroke. To make any applicable conclusions on this topic, more clinical studies should be conducted with an emphasis on finding correlations between various Hp genotypes and relevant SCD outcomes. Although there is no strong evidence that Hp genotype affects stroke incidence or outcomes, this discussion has the potential to extend into Hp therapeutics and targeted gene therapy research. Future research should compare treatment with the Hp 1-1 phenotype with “less ideal” phenotypes. Considering the current technology available, the best option may be to test for Hp phenotypes in blood donors and compare Hp phenotypes (e.g., Hp 1F-1F) theorized to be advantageous with Hp 2-2 for transfusion of patients with SCD.

## Figures and Tables

**Figure 1 genes-13-00144-f001:**
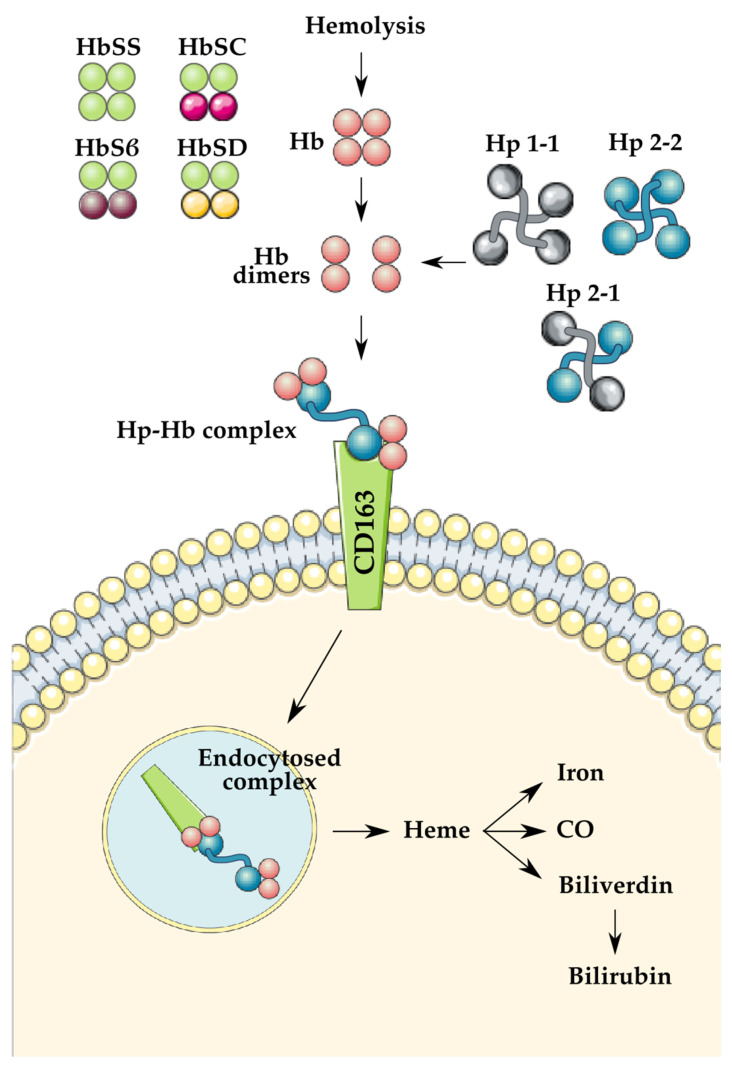
Demonstration of the role of haptoglobin (Hp) in hemoglobin (Hb) clearance following hemolysis. The previously mentioned variants of Hb and Hp are illustrated on the left and the right, respectively. Illustrations used were created by SMART Servier Medical Art under Creative Commons Attribution 3.0 Unported Licensing at https://smart.servier.com/, accessed on 12 November 2021.

**Figure 2 genes-13-00144-f002:**
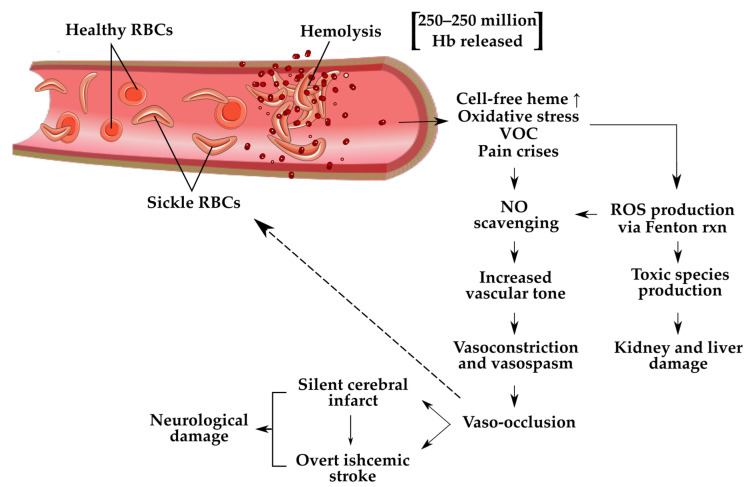
Summarization of hemolytic cascade reactions related to stroke and critical organ damage. Illustrations used were created by SMART Servier Medical Art and modified by the authors under Creative Commons Attribution 3.0 Unported Licensing at https://smart.servier.com/, accessed 12 November 2021.

**Table 1 genes-13-00144-t001:** Summary of clinical studies relating SCD, Hp genotype, and stroke occurrence.

Reference	Descriptive Parameters	Study Design	Anatomical Outcomes	Functional Outcomes
Atkinson et al. [9] Seasonal childhood anaemia in West Africa is associated with the haptoglobin 2-2 genotype	671 Mandinka + Fulani children of age range 2–6 years No chronic illness Jul 2001–Jan 2002 West Kiang, Gambia	Prospective cohort study	- No significant trend observed between the Hp genotype and any sickle cell genotype - Hp genotype was not associated with baseline Hb levels, but children carrying Hp 2-2 showed twice the decrease in Hb over malaria season (*p* = 0.045) as Hp 1-2 children - Glucose-6-phosphate dehydrogenase (G6PD) deficiency was not significantly correlated with Hb drop in children with SCD	- Hp 2-2 was associated with a greater risk for anemia in malaria-infected children
Adekile and Haider [28] Haptoglobin gene polymorphisms in sickle cell disease patients with different βS-globin gene haplotypes	82 Kuwaiti patients with SCD and 49 Kuwaiti control patients 54 South Nigerian patients with SCD and 32 South Nigerian control patients	Prospective case–control study	- Insignificant distribution pattern of Hp genotypes (*p* = 0.78, *p* = 0.41) - Significant (χ² = 31.4, *p* < 0.01) Hp genotype distribution between SCD groups - 52.4% of Kuwaiti patients with SCD were Hp 2-2 and only 16.7% of South Nigerian patients with SCD were Hp 2-2 - 37.5% of South Nigerian patients with SCD were Hp 1-1 and only 4.9% of Kuwaiti patients with SCD were Hp 1-1 - No significant difference among South Nigerian states - Kuwaiti patients with SCD and controls showed almost no difference in HP 2 allele distribution (73.8%, 71.4%) - South Nigerian patients with SCD and controls showed almost no difference in HP 1 allele distribution (60.7%, 54.7%) - No significant difference (*p* = 0.29) was found across Hp allele frequency studywide	- No significant differences were found between HP 2 allele presence and VOC frequency in the Kuwaiti group - Kuwaiti patients with SCD showed frequent VOC and no HP 2 allele association was found (Nigerian patients not stratified) - Stroke was uncommon among the Kuwaiti patients with SCD
Cox et al. [29] Haptoglobin, α-thalassaemia and glucose-6-phosphate dehydrogenase polymorphisms and risk of abnormal transcranial Doppler among patients with sickle cell anaemia in Tanzania	601 Tanzanian patients with SCD <24 years 2004–2005, 2009 and 2010 Excluded blood transfusion in past 2 months, a sickle crisis in the past 2 weeks, history of stroke	Cross-sectional descriptive study	- Of the 601 patients with SCD, 23.46% had Hp 1-1, 56.64% had Hp 1-2, and 19.91% had Hp 2-2 genotypes - No significant difference between Hp 1-1 and Hp 2-2 in terms of CBF (*p* = 0.633) - 2-deletion α-thalassemia was inversely associated with CBF (*p* = 0.002) - Hb levels were inversely associated with CBF (*p* = 0.007)	N/D
Barbosa et al. [8] Haptoglobin and myeloperoxidase (-G463A) gene polymorphisms in Brazilian sickle cell patients with and without secondary iron overload	78 Brazilian patients with homozygous SCA (HbSS) (34 M, 44 F), aged 21–65 years, 59 patients acquired iron overload and 19 patients normal iron, Federal District, Brazil	Cross-sectional study (convenience sample)	- 32% of patients with SCA with acquired iron overload exhibited Hp 2-2 - Hp 1F-1S and Hp 1F-1F were observed in only 15% of the 78 patients with SCA - Hp 2-2 patients with iron overload had a higher platelet count, but those without iron overload presented inconsistencies - No statistically significant distribution of Hp genotypes between iron overload and normal patients with SCA - Patients with iron overload and Hp 2-2 might experience a lesser change in blood flow dynamics due to SCA	- Lower risk of stroke linked to less Hp 1S-2 frequency than predicted (*p* = 0.005) - Hp 1S-2 showed the highest percentage of hospitalization for stroke and sequelae of stroke - Hp 1S-2 showed the highest hospitalization and sequelae of stroke, according to a large deviation from HWE; 21.57 patients predicted, only 10 patients observed - Hp 1F-1F group presented no patients with hospitalization for stroke - Hp 1S-2 and stroke hospitalization showed a significantly high odds ratio (OR = 6.346, *p* = 0.005) similar to Hp 1S-2 and sequelae of stroke (OR= 6.556, *p* = 0.005)
Olatunya et al. [26] Haptoglobin gene polymorphism in patients with sickle cell anemia: findings from a Nigerian cohort study	101 Nigerian patients with stable SCA (67 M, 34 F) of age range 2–21 years, median 9 years SCA patients no crises for 1 mo and no transfusion for 100 days 64 healthy control patients; 40 M,24 F Ado Ekiti, Nigeria	Cross-sectional descriptive study	- No significant genotype distribution found between SCA and control groups - The HP 1 allele was more frequent among patients with SCA (~62%) but less frequent than controls (~73%); it was not statistically different (*p* = 0.06) - Study supports previous northeast Brazilian and Nigerian studies where the phenotype was more severe; genotypes associated with the HP 1 allele were more common among Nigerian patients with SCA	- Stroke identified in 5 patients: 2 Hp 1-1, 1 Hp 2-1, 2 Hp 2-2. Insignificant distribution (*p* = 0.375) - Study contrasts previous studies suggesting that certain SCA patient Hp genotypes (Hp 2-2) influence poor clinical health - Hp genotype is likely not a reliable indicator of clinical health in patients with SCA due to various insignificant results produced by this study

## Data Availability

Not applicable.

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
