# Peer review of "Influence of Haptoglobin Polymorphism on Stroke in Sickle Cell Disease Patients"

_genes, 2022, doi:10.3390/genes13010144_

Round 1

Reviewer 1 Report

Edwards and team provide a comprehensive review on how haptoglobin polymorphisms may be implicated in stroke outcomes in patients with SCD. The paper is well-organized and Table 1 nicely summarizes available literature. I found portions of the article a little cumbersome to read particularly in summarizing the current evidence, but the table was helpful to sift through the data.  It would be interesting to me in the future directions area to include a comment on how voxelotor and crizanlizumab may impact haptoglobin parameters.

Reviewer 2 Report

This review article by O. Edwards et al. aims to identify the influence of the haptoglobin (Hp) genetic polymorphism on clinical manifestations of sickle cell disease (SCD), focusing on its influence in stroke incidence. This is due to the Hp potential to being an efficient treatment for SCD patients. This review summarizes nicely the state of the art for this topic and highlight the lack of specific designed studies to asses more efficiently the association for Hp polymorphisms with the stroke incidence. Nevertheless, I would like to report some minor issues before this article is accepted in Genes journal:

  1. There are some issues with the figures. The legend of Figure 1 refers to the image on Figure 2 and the legend of Figure 2 refers to the image on Figure 1. Thus, according to the text, Figure 1 legend is okay but the image should be the one that is current on Figure 2; and the same happens for Figure 2, the legend is okay but the image should be the one that is on Figure 1.
  2. The font size of the current Figure 1 image is not legible; it should be increased.
  3. Table 1 is not referenced in the text. I think a references should be added at the beginning of section ‘3. Current Evidence Regarding Hp Genotypes and Stroke’, including a phrase saying how many papers were included in this review.
  4. It would be good if the authors discuss why not to do a meta-analysis with all the papers included in this review. A meta-analysis may show more clearly if the polymorphisms of Hp are associated or not with the incidence of stroke in SCA patients.
